# In Vivo Metabolism of [1,6-^13^C_2_]Glucose Reveals Distinct Neuroenergetic Functionality between Mouse Hippocampus and Hypothalamus

**DOI:** 10.3390/metabo11010050

**Published:** 2021-01-12

**Authors:** Antoine Cherix, Rajesh Sonti, Bernard Lanz, Hongxia Lei

**Affiliations:** 1Laboratory of Functional and Metabolic Imaging (LIFMET), Ecole Polytechnique Fédérale de Lausanne, CH-1015 Lausanne, Switzerland; antoine.cherix@ndcn.ox.ac.uk (A.C.); rajesh.sonti@niperhyd.ac.in (R.S.); bernard.lanz@epfl.ch (B.L.); 2Animal Imaging and Technology (AIT), Center for Biomedical Imaging (CIBM), Ecole Polytechnique Fédérale de Lausanne, CH-1015 Lausanne, Switzerland; 3Faculty of Medicine, University of Geneva, CH-1206 Geneva, Switzerland

**Keywords:** brain metabolism, in vivo imaging, mathematical modeling, metabolic flux analysis, ^13^C-MRS

## Abstract

Glucose is a major energy fuel for the brain, however, less is known about specificities of its metabolism in distinct cerebral areas. Here we examined the regional differences in glucose utilization between the hypothalamus and hippocampus using in vivo indirect ^13^C magnetic resonance spectroscopy (^1^H-[^13^C]-MRS) upon infusion of [1,6-^13^C_2_]glucose. Using a metabolic flux analysis with a 1-compartment mathematical model of brain metabolism, we report that compared to hippocampus, hypothalamus shows higher levels of aerobic glycolysis associated with a marked gamma-aminobutyric acid-ergic (GABAergic) and astrocytic metabolic dependence. In addition, our analysis suggests a higher rate of ATP production in hypothalamus that is accompanied by an excess of cytosolic nicotinamide adenine dinucleotide (NADH) production that does not fuel mitochondria via the malate-aspartate shuttle (MAS). In conclusion, our results reveal significant metabolic differences, which might be attributable to respective cell populations or functional features of both structures.

## 1. Introduction

The brain is well known to be a particularly energy-demanding organ [1], however, less is known about its regional metabolic specificities and energetic requirements. While different brain structures are involved in distinct information processing circuits and behaviors, their energy metabolism is likely to reflect their structure and function, leading to region-specific metabolic characteristics [2,3]. Hippocampus and hypothalamus are both fundamental brain structures that are part of the limbic system, involved in the regulation of metabolism and behavior, and thus critically important in the response to stress [4,5]. Both structures are known to mediate distinct tasks, however, less is known about how this translates into differences in their basal energy requirements. Hypothalamus is a deep brain structure that contains different hormone- or energy metabolite-sensing neuronal populations [6]. As such, hypothalamus is a key central regulator of body metabolism and energy balance and is therefore particularly affected as a result of metabolic disorders [7]. Hippocampus, on the other hand, has been implicated with more higher-order information processing, including memory formation, learning and goal directed behavior [8,9,10]. Interestingly, hippocampus is a well-known target of excessive stress exposure, where high cellular energy demands is thought to induce neuronal damage and cell death [11,12].

Magnetic resonance spectroscopy (MRS) provides an outstanding way of studying metabolism in vivo allowing to obtain information from different brain structures non-invasively [13,14,15,16]. In particular, carbon-13 (^13^C) MRS allows the investigation of metabolism with cell-specific information through dynamic assessment of metabolic fluxes in vivo upon administration of a ^13^C-labeled substrate [17,18]. We have previously shown that studying region-specific brain metabolism of mouse was feasible at ultra-high field using indirect carbon-13 MRS (^1^H-[^13^C]-MRS) with ^13^C-labeled substrates such as [1,6-^13^C_2_]glucose ([1,6-^13^C_2_]Glc) [16,19] or [U-^13^C_6_]glucose ([U-^13^C_6_]Glc) [20]. Because variations in the acquisition protocol or mathematical modeling of brain metabolic fluxes can lead to slight variations in the results that render region-to-region statistical comparisons challenging, we sought of evaluating hippocampus and hypothalamus metabolism under similar experimental and analytical conditions. Besides allowing a comparison with previous work that looked at both structures separately, hippocampus and hypothalamus were selected due to their distinct roles within the limbic system and their known sensitivity to metabolic challenges.

In this study, we have investigated for in vivo differences in the energy metabolic function of hippocampus and hypothalamus in mice using ^1^H-[^13^C]-MRS upon infusion of [1,6-^13^C_2_]Glc. By combining an updated mathematical model of brain metabolism that includes GABAergic contribution and cerebral metabolic rate of glucose (CMR_Glc_), we have been able to assess simultaneously glycolytic and mitochondrial contributions non-invasively. We report that hypothalamus has higher rates of ATP synthesis and undergoes higher levels of aerobic glycolysis than hippocampus, possibly related to its glucose sensing properties. Better characterizing and understanding region-specific metabolic properties will help further our understanding on how metabolic diseases or stress may affect cerebral substructures differently.

## 2. Results

### 2.1. ^1^H-[^13^C]-MRS Data Comparison between Hypothalamus and Hippocampus at 14.1T

The non-edited spectra were of similar quality, with a SNR of 20 ± 2 for hippocampus and 16 ± 3 for hypothalamus, when adjusting the time resolution to ~5.5 min and ~11 min for each region, respectively, to compensate for the lower signal in the hypothalamic voxel (Figure 1). The resulting metabolic linewidth, as calculated by LCModel, was 18 ± 2 Hz for hippocampus and 17 ± 2 Hz for hypothalamus. The edited spectra were comparable as well between the two structures, as indicated by the similar CRLBs of most metabolite resonances, even though the SNRs of hippocampal spectra remained slightly higher over time (Appendix A). The edited spectra timeline of both structures indicated a rapid increase in GlcC6 and LacC3 labeling from the first time points, while remaining ^13^C-coupled ^1^H resonances such as GluC4, GlnC4, GlxC3, GlxC2, GABAC2 and GABAC3 were all clearly observed after ~1 h of [1,6-^13^C_2_]Glc infusion (Figure 1c,d). Reliable FE (using quantifications with CRLB < 50%) were calculated for GlcC6, LacC3, GluC4, GlnC4, GlxC3, GlxC2, GABAC2 and GABAC3, with steady-state values summarized in Appendix A.

### 2.2. Metabolic Differences in Neurochemical Profile between Hippocampus and Hypothalamus

The neurochemical profile was quantified from the non-edited spectra, as described in the methods, and revealed important metabolic differences between hypothalamus and hippocampus (Figure 2). Energy-related metabolites such as Cr (+32%, *p* < 0.0001), Tau (+57%, *p* < 0.0001) and Asp (+40%, *p* = 0.027) were found to be higher, while Lac (−24%, *p* = 0.049) was lower in hippocampus as compared to hypothalamus. Antioxidant Asc level was higher (+121%, *p* = 0.0008) in hippocampus, while GSH was comparable (n.s.). Importantly, neurotransmitter metabolites Glu and Gln were not found to be different (n.s.), while GABA levels were lower (−39%, *p* = 0.0002) in hippocampus. The membrane-degradation metabolite GPC was lower in hippocampus (−72%, *p* = 0.0024), while no difference in membrane-precursor PCho was seen (n.s.). Finally, the glia-specific myo-Ins metabolite was lower in hippocampus (−13%, *p* = 0.0046).

### 2.3. Mathematical Modeling of Hippocampal and Hypothalamic Metabolism

The fitting of metabolite enrichment time courses with a 1-compartment model of glucose metabolism (Figure 3) led to comparable results between both brain regions (hypothalamus, R^2^ = 0.973; hippocampus, R^2^ = 0.994) and is shown in Figure 4. The metabolic fluxes estimated from this mathematical analysis are summarized in Table 1 and reveal some region specificities described as follows. Firstly, the cerebral metabolic rate of glucose (CMR_Glc_; hypo: 0.47 ± 0.06 vs. hippo: 0.29 ± 0.02 μmol/g/min; *p* < 0.0001) was found to be higher in hypothalamus than hippocampus, while tricarboxylic acid (TCA) cycle (V_TCA_; hypo: 0.87 ± 0.11 vs. hippo: 0.75 ± 0.05 μmol/g/min; n.s.) and neurotransmission flux were comparable (V_NT_; hypo: 0.14 ± 0.05 vs. hippo: 0.14 ± 0.01 μmol/g/min; n.s.). Accordingly, the relative higher glycolytic activity of hypothalamus led to an apparent increase in lactate efflux (V_dil_^out^; hypo: 0.30 ± 0.02 vs. hippo: 0.21 ± 0.02 μmol/g/min; *p* < 0.0001) and a decreased influx (V_dil_^in^; hypo: 0.00 ± 0.06 vs. hippo: 0.23 ± 0.06 μmol/g/min; *p* < 0.0001) from blood lactate. Although total mitochondrial oxidative metabolism (i.e., V_TCA_ + V_PC_ + V_GABA_) appeared to be lower in hippocampus (−16%), the transmitochondrial flux (V_x_; hypo: 0.20 ± 0.06 vs. hippo: 0.50 ± 0.08 μmol/g/min; *p* < 0.0001), reflecting malate-aspartate shuttle (MAS) flux, and glial AcCoA dilution flux (V_dil_^g^; hypo: 0.00 ± 0.00 vs. hippo: 0.06 ± 0.01 μmol/g/min; *p* < 0.0001), associated with ketogenic substrate uptake and glial dilution, were both increased in this structure as compared to hypothalamus. Nevertheless, this did not appear to be directly related to a difference in glial metabolism as the pyruvate carboxylase flux was low in both regions (V_PC_; hypo: 0.010 ± 0.010 vs. hippo: 0.005 ± 0.005 μmol/g/min; n.s.). As expected from the higher GABAergic neuron population of hypothalamus, the GABA-Glu cycle (V_GABA_; hypo: 0.09 ± 0.02 vs. hippo: 0.06 ± 0.01μmol/g/min; *p* < 0.0001) was found higher in this structure, but not the GABA dilution flux (V_ex_^i^; hypo: 0.02 ± 0.02 vs. hippo: 0.01 ± 0.01 μmol/g/min; n.s.).

## 3. Discussion

In this study, we present a new comparative energetic analysis of hippocampus and hypothalamus that considers both glycolytic and oxidative metabolism in vivo using ^1^H-[^13^C]-MRS upon infusion of [1,6-^13^C_2_]Glc. By comparing both regions during a similar experimental protocol and using a simple model of brain metabolism that considers GABAergic contribution with minimal assumptions, we have been able to estimate the cerebral metabolic rate of glucose without addition of ^18^F-fluorodeoxyglucose positron emission tomography (^18^FDG-PET) measurement. We report that under our experimental conditions, the hypothalamus shows a higher rate of glycolytic activity, as compared to the hippocampus that appears to rely more on oxidative metabolism (Figure 5).

### 3.1. Protocol Optimization Allows Simultaneous Assessment of Glycolytic and Oxidative Glucose Metabolism

We have previously reported that estimating brain metabolism with a model that includes GABAergic metabolic contribution is feasible in the hypothalamus [19], however, assessing glycolytic pathway of [1,6-^13^C_2_]Glc metabolism remained to be achieved. As such, by using additional information from plasma measurements, with interpolation of minimal time-points, we have been able to assess CMR_Glc_, V_dil_^in^ and V_dil_^out^. We have previously reported that the inclusion of a dilution flux (V_ex_^i^), that corresponds to the labeling exchange between two distinct GABA pools, provides a more accurate means of fitting the GABA labeling curves [20]. As such, inclusion of this flux in the GABAergic model would allow us to assess hypothalamic GABAergic metabolism in a more detailed way than previously described [19]. Although the model used herein remains comparable to that presented in Cherix et al. [20], the determination of CMR_Glc_ independently of PET measurement had not been tested before. As compared to previous studies that have assessed metabolism in hypothalamus and hippocampus separately [19,20], we have restricted our analysis to the use of a single compartment mathematical model to reduce the number of estimated parameters and focus on the overall regional differences to evaluate glycolytic and oxidative metabolism simultaneously. Importantly, using a similar protocol (infusion substrate, infusion duration, mathematical model) allowed us to directly compare metabolic fluxes between both regions.

### 3.2. Major Metabolic Differences with Current Literature

For the first time, we report an age- and protocol-matched ^1^H-[^13^C]-MRS study that compares neuroenergetics of hippocampus and hypothalamus in vivo. The distinct neurochemical concentration characteristics (Figure 2) were largely similar to previously reported hippocampus and hypothalamus quantifications [7,21,22]. Furthermore, metabolic fluxes were overall comparable to previous studies using ^1^H-[^13^C]-MRS but present some discrepancies that might arise from the differences in protocol or modeling strategy.

We previously described [1,6-^13^C_2_]Glc metabolism in the hypothalamus of mice using a similar infusion protocol [19]. Although most of the fluxes are in good agreement with this study, we report significantly lower transmitochondrial flux (V_x_; 0.68 ± 0.21 [19] vs. 0.20 ± 0.06 μmol/g/min here), and neurotransmitter cycling (V_NT_; 0.41 ± 0.07 [19] vs. 0.14 ± 0.05 μmol/g/min here). Several factors might have contributed to these major differences. As we did not include here the labeling curves of AspC3, due to unreliable quantification, this might have impacted the estimation of V_x_. Nevertheless, while we previously studied hypothalamic metabolism using a 190 min infusion protocol, the 240 min protocol used here might have allowed a better assessment of the enrichment plateau leading to a more reliable flux estimation, and/or impacted the neuronal neurotransmission through prolonged exposure to isoflurane, known to reduce V_NT_ [23,24]. Another source of variation might be the difference in modeling strategy, as Lizarbe et al. [19] did not consider metabolism upstream of mitochondria and limited the analysis to the use of LacC3 labeling as an input function.

We have also previously reported hippocampal glucose metabolism in vivo using, however, a different labeled substrate ([U-^13^C_6_]Glc). Although the transmitochondrial flux was comparable (V_x_; 0.48 ± 0.26 [20] vs. 0.50 ± 0.08 μmol/g/min here), the neurotransmission flux of hippocampus was found to be ~2-fold higher here (V_NT_; 0.06 ± 0.01 [20] vs. 0.14 ± 0.01 μmol/g/min here). Interestingly, the infusion duration cannot be accountable for this difference and other factors such as animal age (6 weeks [20] vs. 8–12 weeks here), daytime or physiological parameters might have influenced this result. Mostly, differences arise in the TCA (V_TCA_; 1.71 ± 0.03 [20] vs. 0.75 ± 0.05 μmol/g/min here) and glycolysis (CMR_Glc_; 0.61 ± 0.02 [20] vs. 0.29 ± 0.02 μmol/g/min here) fluxes, as Cherix et al. [20] included values obtained from a PET measurement, which reflects more accurately brain metabolism under euglycemia. Of note, we did not assess the GABAergic dilution flux in a single-compartment model previously, but the present result suggests no difference between hypothalamus and hippocampus (hippo: 0.01 ± 0.01 vs. hypo: 0.02 ± 0.02 μmol/g/min, n.s.) and the absolute values were higher than those found with a pseudo 3-compartment model (i.e., ~20-fold higher) [20].

### 3.3. Biological Functionality Underlying Regional Differences in Energy Metabolism 

The findings of our cross-regional comparison provide important insights in the respective metabolic activities of hypothalamus and hippocampus. In particular, our results suggest that hypothalamus undergoes significantly more aerobic glycolysis than hippocampus for a similar neurotransmission activity (Figure 5). We have recently reported that TCA cycle activity relative to neurotransmission in mouse hippocampus is high [20] as compared to whole brain [16,25] and hypothalamus [19]. Nevertheless, the different ^13^C-MRS protocols and analysis pipelines make the cross-region comparisons between studies difficult. Although oxidative metabolism (CMR_Glc_(ox)) was found to be slightly higher in hypothalamus compared to hippocampus here, we observed higher mitochondria-related metabolites (Cr, Tau and Asc) and transmitochondrial flux (V_x_) in the hippocampus, suggesting an important mitochondrial dependence for energy metabolism. Little is known about region-specificities in mitochondrial function, and only mitochondrial mass has been compared between hippocampus and hypothalamus, suggesting higher content in the latter [26]. Recently, Andersen et al. [27] have observed comparable efficiency (i.e., ATP synthesis/O_2_ consumed) of isolated non-synaptic mitochondria between hippocampus and striatal or cortical regions. Nevertheless, the authors reported differences in basal and coupled oxygen consumption rates, suggesting hippocampus produces ATP at lower rates than cortex and striatum [27]. Our ^1^H-[^13^C]-MRS protocol does not allow the quantification of ATP synthesis rate, however, by assuming a similar ATP yield for both structures (i.e., ~23 ATP/pyruvate through PDH-TCA pathway and ~2.5 ATP/NADH through MAS) [28], our results indicate a −14% lower ATP production rate in hippocampus as compared to hypothalamus (hippo: 20.5 vs. hypo: 23.7 μmol ATP/g/min; Appendix A). These calculations assume that the V_x_ flux reflects the transamination process involved in the MAS, which we found to be 2-fold higher in hippocampus compared to hypothalamus, and that coupling between electron transport chain (ETC) and ATP-synthase is comparable between structures. The MAS allows to shuttle the excess NADH produced in the cytoplasm to be further oxidized by the ETC of the mitochondria, and has been considered as a major metabolite shuttle for neurons in the brain [29]. Interestingly, from our calculations, it appears that cytosolic NADH production rate is ~4-fold higher in hypothalamus (Appendix A), as a consequence of higher glycolytic activity. Accordingly, high level of cytoplasmic NADH production coincide with higher levels of lactate measured in the hypothalamus, in line with the idea that recycling of NAD^+^ with lactate dehydrogenase (LDH) helps avoiding glycolytic inhibition. High production levels of reducing equivalents are important for biosynthetic purpose, and production of nicotinamide adenine dinucleotide (NADPH) from NADH, involving successive activity of pyruvate carboxylase (PC), malate dehydrogenase (MDH) and malic enzyme (ME), has been described in the brain and could reflect astrocytic metabolism [30]. Furthermore, higher reducing equivalent production rates in hypothalamus could potentially explain the relatively low levels of ascorbate antioxidant concentration in that structure (but similar GSH levels) compared to hippocampus. Of note, our analysis does not assess the amount of NADH transferred for mitochondrial oxidation through the glycerol-3-phosphate shuttle [29], and further experiments would be required to determine whether hypothalamus relies more specifically on this pathway for oxidizing cytosolic NADH.

Several cytoarchitecture or functional characteristics might explain the observed differences in glucose metabolism between hippocampus and hypothalamus. For instance, high glial and GABAergic densities in the hypothalamus might explain the higher rates of aerobic glycolysis in this structure. The glia/neuron density ratio has been reported to be higher in hypothalamus (0.79 ± 0.12) compared to hippocampus (0.54 ± 0.17) [31]. Astrocytes have been reported to undergo high levels of glycolysis and lactate release in the adult rodent brain, providing trophic/metabolic support for adjacent neurons [32]. Along with this hypothesis of higher glial metabolism, we observed an increase in glia-related [33] membrane metabolites (GPC, myo-Ins) and a slight, yet not statistically significant, increase in pyruvate carboxylase flux (hippo: 0.005 ± 0.005 vs. hypo: 0.010 ± 0.010 μmol/g/min) in hypothalamus as compared to hippocampus. Nevertheless, it is also important to note that the relative excitatory/inhibitory neuron density ratio was reported to be overall lower in hypothalamus (3.51 ± 0.57) as compared to hippocampus (12.4 ± 4.0) [31] and could thus contribute to the difference in energy metabolism organization as well. Accordingly, GABA concentration and V_GABA_ were found to be ~60% and ~50% higher, respectively, for hypothalamus in this study, in line with the high GABAergic neuron population density in hypothalamus. Finally, a key explanation for these regional neuroenergetic differences might lie in the well-established glucose-sensing properties of hypothalamus [34]. The high expression and variety of glucose transporters in hypothalamus [35] could thus underlie the observed levels of aerobic glycolysis.

### 3.4. General Concerns about the Protocol and Anesthesia

Several factors are likely to influence brain metabolic fluxes, such as differences in the protocol (e.g., rodent model, infusion substrate, infusion time, etc.) and analysis (e.g., mathematical model, prior knowledge, assumptions, etc.). However, our approach aimed at minimizing these effects by comparing both structures under similar experimental and analytical conditions. Nevertheless, anesthetics, and in particular isoflurane are known to affect neuronal electrical and metabolic activity, and the resulting metabolic state observed, even though studied in similar conditions, might be influenced by the respective neuronal/glial populations. For instance, we cannot exclude that the high level of glycolytic activity in hypothalamus is merely a consequence of a higher impact of isoflurane on its metabolic/electric activity. Nevertheless, this potential “differential response” to isoflurane between brain structures might by itself provide important metabolic functional information. We conclude that ^1^H-[^13^C]-MRS at 14.1T upon infusion of [1,6-^13^C_2_]Glc enables detection of region-specific differences in mitochondrial and glycolytic metabolism in vivo. Comparison of the hippocampus and hypothalamus metabolism under similar experimental and analytical conditions allowed to identify a higher level of aerobic glycolysis in the hypothalamus, which might be associated to its particular glucose sensing properties and/or specific cellular population densities.

## 4. Material and Methods

### 4.1. Animals

All experiments were approved by the local ethics committee (Service de la consommation et des affaires vétérinaires, Epalinges, Switzerland) under license VD3296 and performed according to the ARRIVE (Animal Research: Reporting of In Vivo Experiments) guidelines. One week after arrival, 14 adult C57BL/6J male mice (Charles Rivers Laboratories, L’Arbresle, France; 8–12 weeks old, 25 ± 2 g) were set up for magnetic resonance spectroscopy (MRS) and imaging (MRI) experiments to assess either the hippocampus (*n* = 8) or the hypothalamus (*n* = 6) after a 10h-long fasting period (glycemia at 7 ± 1 mM). First, a femoral vein was cannulated for infusion of a 20% glucose solution of [1,6-^13^C_2_]Glc (Sigma-Aldrich, St. Louis, MO, USA). This solution was first administered as a bolus (99% ^13^C-enriched glucose, 9 mL/kg during 5 min at an exponential decay rate), and secondly at a constant infusion rate (62% enrichment substrate, 10 mL/kg/h) during the rest of the experiment [36]. All MRS measurements were performed with animals under isoflurane anesthesia (3–4% for induction, 1%–1.5% for maintenance).

### 4.2. MR Hardware

All measurements were carried out in a horizontal 26-cm-diameter 14.1Tesla magnet (Magnex Scientific, Abingdon, UK), equipped with a 12-cm internal diameter gradient coil insert (400 mT/m, 120 μs), and interfaced to a DirectDrive console (Agilent Technologies Inc., Santa Clara, CA, USA). A home-built radiofrequency (RF) coil was optimized for ^1^H-[^13^C]-MRS studies on mouse brain [20]. In particular, a quadrature ^1^H coil (two geometrically decoupled 10 mm-inner-diameter loops, <15 dB) was shaped in a half volume covering the entire mouse head and further decoupled with an additional single 8 mm-inner-diameter loop ^13^C coil (two-turned 1 mm-diameter enameled copper wire, Part 1230985, Rowan Cable Products Ltd., Potters Bar, UK), i.e.,<−30 dB.

### 4.3. MR Method

Anatomical MR images were acquired with fast spin-echo sequence (TE_effective_/TR = 12.3/2000 ms repetition time, echo train length = 4, averages = 1, 30 × 0.4 mm slices, field of view = 20 × 15 mm^2^, data matrix = 128 × 64). In this study, the volume of interest (VOI) was either the bilateral dorsal hippocampus (6 × 1.8 × 2 mm^3^) or the hypothalamus (2.7 × 2 × 2.2 mm^3^), in which the field homogeneity was improved by adjusting both first and second order shim gradients using FASTMAP methods [37]. A B_1_-insensitive spectral editing pulse (BISEP) in the ^1^H channel in combination with an adiabatic ^13^C inversion pulse OFF and ON in an interleaved mode was implemented within a localized MR spectroscopy sequence, i.e., SPECIAL [38], as previously described in Xin et al. [39]. In addition, outer volume suppression (OVS) was applied and water suppression was performed using a VAPOR scheme with seven chemical shift selective pulses and an additional 12-ms Gaussian CHESS pulse between the spatially selective adiabatic pulse and the excitation pulse. In the BISEP module, the bandwidth of the inversion pulse at γB_1max_/2π of ^1^H channel was ~2 kHz (3.3 ppm at 14.1 T), and the bandwidth of the inversion pulse of ^13^C channel 12 kHz (80 ppm at 14.1 T) when γB_2max_/2π = 7 kHz. Finally, a decoupling bandwidth of 9 kHz (60 ppm at 14.1 T) was achieved in the ^13^C channel using hyperbolic secant HF8 adiabatic full-passage pulse during the entire acquisition period.

### 4.4. ^1^H-[^13^C]-MRS Spectra Analysis

All spectral data were collectively saved and processed offline. All spectra were frequency corrected and summed/subtracted in the desired number of scans, i.e., 80 and 160 for hippocampus and hypothalamus respectively. Quantification of metabolites was performed on all spectra using LCModel [40]. For instance, the non-edited ^1^H MR spectra were quantified with a standard basis set containing the measured macromolecule and simulated metabolite spectra, including alanine (Ala), aspartate (Asp), phosphocholine (PCho), creatine (Cr), phosphocreatine (PCr), γ-aminobutyric acid (GABA), glutamine (Gln), glutamate (Glu), glutathione (GSH), glycine (Gly), myoinositol (myo-Ins), lactate (Lac), *N*-acetylaspartate (NAA), scyllo-inositol (scyllo-Ins), taurine (Tau), ascorbate (Asc), glucose (Glc), *N*-acetylaspartyl glutamate (NAAG), glycerophosphorylcholine (GPC), phosphoethanolamine (PE) and acetate. The ^13^C-edited spectra were quantified using another simulated basis set containing the resonances of ^1^H coupled to NAA C6, Glu (C2, C3, and C4), Gln (C2, C3, and C4), GABA (C2, C3, and C4), Asp (C2 and C3), Glc (C1-C6), Cr+PCr (C2 and C3), Lac C3, and Ala C3. Total pool size of metabolites was calculated using water (80% in the brain tissue, 44.4 mol/L) as an internal reference, and the FE of GluC6, LacC3, GluC4, GlnC4, Glx (C3,C2) and GABA (C2, C3) were calculated for every time point and animal. Finally, the FE were multiplied by the associated metabolite concentration to obtain the ^13^C-labeling curves (mean ± SD) used for the metabolic flux analyses after averaging them for all animals.

### 4.5. FE Measurements of Blood Samples

Plasma samples (*n* = 10) collected randomly from mice immediately after in vivo experiments were stored at −80 °C until analyzed by NMR. After thawing the frozen samples, a 50 μL aliquot of plasma was dissolved in PBS buffer, pH adjusted to 7.4 for a final concentration of a solution having 0.8 mol/L ammonium chloride (NH_4_Cl), 2 mmol/L sodium formate and 10% D_2_O. Use of 0.8 mol/L NH_4_Cl resulted in complete water suppression and the release of protein-bound metabolites. All NMR experiments were recorded at 400 MHz on a Bruker spectrometer (Bruker BioSpin, Ettlingen, Germany) equipped with a BBFO probe at 298 K using 5 mm tubes (Wilmad-LabGlass, Vineland, NJ, USA). Sodium formate (analytical standard, Sigma-Aldrich Inc., St. Louis, MO, USA) was used as internal standard for chemical shift referencing and integration. A cpmgpr pulse program with an echo delay of 250 μs and a loop counter of 1400 was used, which resulted in the complete removal of the contribution of macromolecular species to the final spectrum. All spectra were recorded with 16 scans having a 4.8 kHz spectral width with a presaturation time of 3 s and a repetition time of 37 s, that ensured larger than 5*T_1_ of the formate species. The formate singlet peak at 8.47 ppm and the methyl doublet peak of lactate at 1.34 ppm were used for integration. Both T_2_ relaxation times of the formate and lactate peaks were calculated using 10 data points varying from 0.001 to 3.2 s. The measured peak areas from the sample spectra were used as an input for the T_2_ exponential functions of lactate and formate to back calculate the original peak areas. The final lactate concentration was estimated from the standard formate using the equation:(1)$$C_{M}=\frac{C_{F}\times \;N_{F}\times I_{M}}{I_{F}\times N_{M}}$$
where *C_M_* is the concentration of lactate, *C_F_* is the concentration of formate, *N_M_* is the number of protons of lactate, *N_F_* is the number of protons of formate, *I_M_* is the area of lactate peak and *I_F_* is the area of formate peak in the spectra. To measure the intensity of ^13^C lactate proton peak, only the peak at 1.18 ppm was considered, since our pilot results from plasma samples without any labeled infusion of metabolites showed another chemical species at 1.5 ppm having a perfect overlap with the other ^13^C lactate proton peak, and may lead to a wrong estimation of the ^13^C lactate concentration. Using this method, a Lactate concentration of 6.3 ± 1.4 mM and a Lac C3 plasma FE of 0.50 ± 0.03 were obtained and subsequently included in the modeling.

### 4.6. Mathematical Modeling of Metabolic Fluxes

A single-compartment model of brain glucose metabolism was adapted from Lizarbe et al. [19] and modified with additional features from Cherix et al. [20]. In brief, [1,6-^13^C_2_]Glc infusion leads to pyruvate C3 labeling from glycolysis in the brain (CMR_Glc_) and enters mitochondrial TCA cycle (V_TCA_), leading to aspartate and glutamate labeling through the transmitochondrial flux (V_x_). In parallel, the absence of pyruvate C2 labeling, specific to the use of [1,6-^13^C_2_]Glc, dilutes the C3 position of oxaloacetate (OAA), through the action of glia-specific pyruvate carboxylase (V_PC_). Consequently, anaplerotic balance is maintained with an efflux of extra glutamine from glial cells (V_eff_). After neurotransmitter release, glutamate is recycled in astrocytes by conversion into glutamine and recycled back into neurons for replenishing glutamate (V_NT_). In inhibitory neurons, GABA is synthesized from glutamate and recycled through the TCA cycle (V_GABA_). Dilution of GABA occurs in inhibitory neurons through the exchange between two metabolite pools attributed to the effect of both GAD isoforms (V_ex_^i^). Brain lactate is in exchange with plasma lactate through an inward flux (V_dil_^in^) and outward flux (V_dil_^out^), while AcCoA labeling is influenced by plasma acetate and other dilution processes (V_dil_^g^). A schematic description of the model is presented on Figure 3 and mathematical equations are presented in the Appendix A. To estimate hypothalamic and hippocampal metabolic fluxes, the measured ^13^C-labeling curves, i.e., GlcC6, LacC3, GluC4, GlnC4, GlxC3 (i.e., GluC3 + GlnC3), GlxC2 (i.e., GluC2 + GlnC2), GABAC2 and GABAC3, were fitted in the proposed model with MATLAB (Version 8.3, The MathWorks, Inc., Natick, MA, USA) using a standard built-in ordinary differential equation solver with a modified Levenberg-Marquardt nonlinear weighed regression method. GlxC2 labeling curves were corrected to account for their high correlated quantification with GlcC6, as described in Lizarbe et al. [19]. Glucose FE curves were used as an input function for both brain regions and were fitted with an inverse exponential function with oblique asymptote FE = (a·t + b)·(1 − exp(−c·t)). CRLBs of each ^13^C metabolite resonance were used for weighing in the regression cost function. Monte-Carlo simulations (300 artificial curves) were used to evaluate the fluxes precision and providing a probability distribution [17].

### 4.7. Statistics

Differences in metabolic profile of both regions were assessed using unpaired Student’s *t*-test, with a significance threshold set at *p* < 0.05 and no correction for multiple comparison. Group comparisons between metabolite fluxes of the two regions was done using a permutation test analysis with 2000 random permutations, followed by individual unpaired two-tailed Student’s *t*-test, as previously described [41]. Statistical analyses were performed with GraphPad Prism (GraphPad Software, San Diego, CA, USA). All values are given as mean ± SD.

## Figures and Tables

**Figure 1 metabolites-11-00050-f001:**
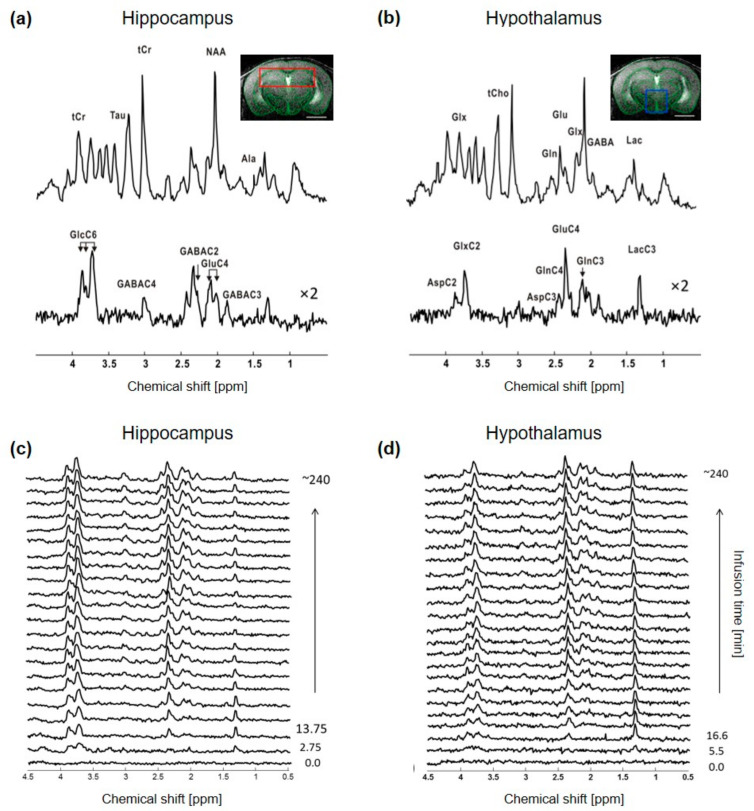
In vivo ^1^H-[^13^C]-MRS acquisition in mouse hippocampus and hypothalamus. Typical ^1^H-[^13^C]-MRS spectra of (**a**) hippocampus and (**b**) hypothalamus of a mouse. The non-edited (top) and edited (bottom) spectra are presented with a Lorentzian apodization of 1 Hz for hippocampus and 2 Hz for hypothalamus and the volume of interest (VOI) is shown in the anatomical image for hippocampus (red, (**a**) panel) and hypothalamus (blue, (**b**) panel). Peak labeling was distributed between the two spectra to avoid over-crowding. Abbreviations: Ala, alanine; Asp, aspartate; Gln, glutamine; Glu, glutamate; Glx, glutamine + glutamate; Tau, taurine; tCr, total creatine; GABA, γ-aminobutyric acid; Lac, lactate; Glc, glucose. Proton resonances bound to specific carbon are indicated by C and followed by the position number, e.g., C2, C3 and C4 etc. Timeline of typical edited spectra of (**c**) hippocampus and (**d**) hypothalamus showing glucose ^13^C labeling incorporation to its downstream brain metabolites throughout the infusion time (minutes). In (**c**), every other edited ^1^H-[^13^C]-MRS spectra were displayed. All spectra are shown with 5 Hz Lorentzian apodization.

**Figure 2 metabolites-11-00050-f002:**
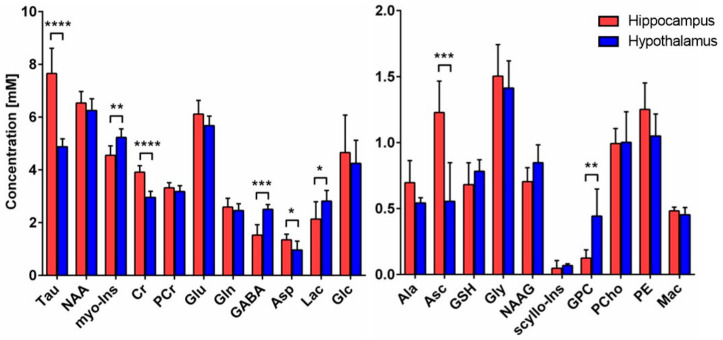
Neurochemical differences between hippocampus and hypothalamus. Neurochemical differences between hippocampus (*n* = 8) and hypothalamus (*n* = 6) were assessed using Student *t*-test; * *p* < 0.05; ** *p* < 0.01; *** *p* < 0.001; **** *p* < 0.0001. Data are presented as mean ± SD. Abbreviations: Tau, taurine; NAA, N-acetyl aspartate; myo-Ins, myo-inositol; Cr, creatine; PCr, phosphocreatine, Glu glutamate; Gln, glutamine; GABA, γ-aminobutyric acid; Asp, aspartate; Lac, lactate; Glc, glucose; Ala, alanine; Asc, ascorbate; GSH, glutathione; Gly, glycine; NAAG, N-acetylaspartyl glutamate; scyllo-Ins, scyllo-inositol; GPC, glycerophosphorylcholine; PCho, phosphocholine; PE, phosphoethanolamine; Mac, macromolecules.

**Figure 3 metabolites-11-00050-f003:**
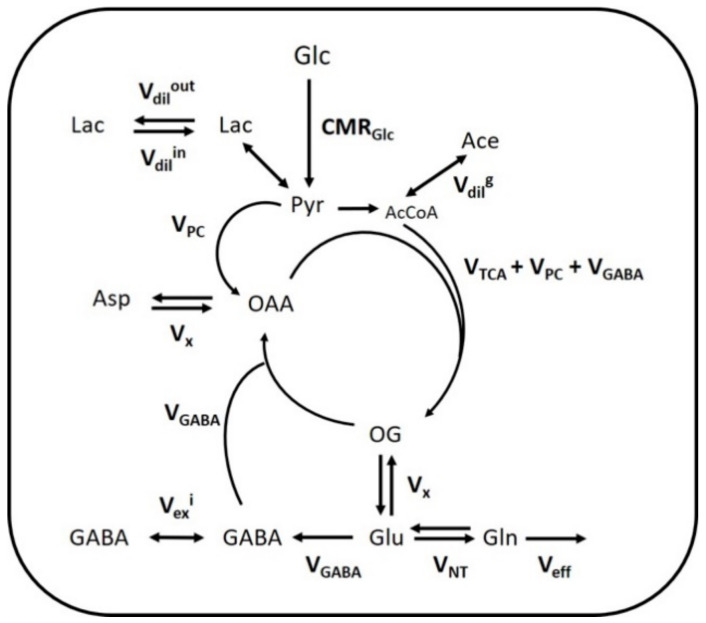
Mathematical 1-compartment model of brain metabolism assessed with [1,6-^13^C_2_]Glc. Labeled and non-labeled glucose (Glc) is metabolized to pyruvate (Pyr) in the brain, which corresponds to the cerebral metabolic rate of glucose (CMR_Glc_). Pyruvate, in fast exchange with lactate (Lac), can either exchange with blood lactate (via V_dil_^in^ or V_dil_^out^), be carboxylated into oxaloacetate (OAA) in glial cells (via V_PC_) or enter mitochondrial tricarboxylic acid (TCA) cycle (via V_TCA_). Dilution can occur in the acetyl-CoA (AcCoA) pool produced from pyruvate, by exchange with ketone bodies, acetate (Ace) or glial-specific metabolism (via V_dil_^g^). Transamination in the mitochondria (V_x_) lead to the exchange of labeling from OAA and oxoglutarate (OG) with aspartate (Asp) and glutamate (Glu) respectively. Glu labels glutamine (Gln) through the neurotransmitter cycling flux (V_NT_), and the excess Gln is released in the blood (V_eff_) to maintain anaplerotic balance. In inhibitory neurons, Glu-GABA flux (V_GABA_) corresponds to the synthesis of GABA from Glu and its recycling into the TCA cycle. Due to potential compartmentation of GABA metabolic pools, an exchange flux can occur, leading to GABA labeling dilution (V_ex_^i^).

**Figure 4 metabolites-11-00050-f004:**
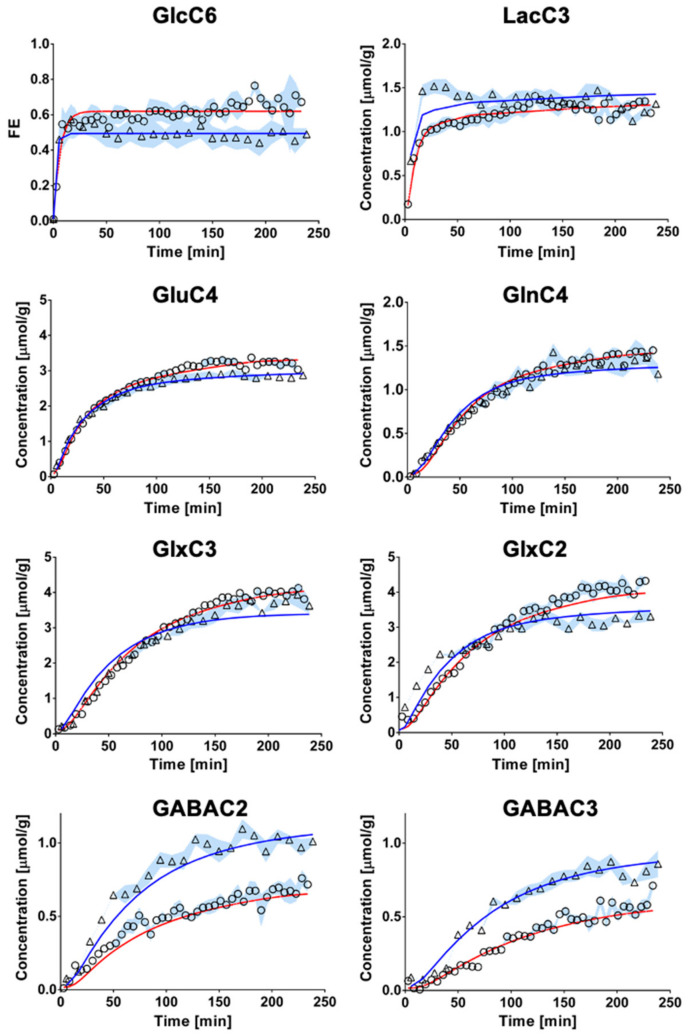
Fitting of the 1-compartment model of brain metabolism to hippocampal and hypothalamic metabolic ^13^C-labeling curves. Hippocampal (circles) and hypothalamic (triangle) ^13^C-labeling curves (mean ± SD) were analyzed with a 1-compartment model of brain metabolism. The resulting overall fit of hippocampus (red line) was comparable to that of hypothalamus (blue line) in terms of goodness-of-fit (hippocampus, R^2^= 0.994; hypothalamus, R^2^ = 0.973).

**Figure 5 metabolites-11-00050-f005:**
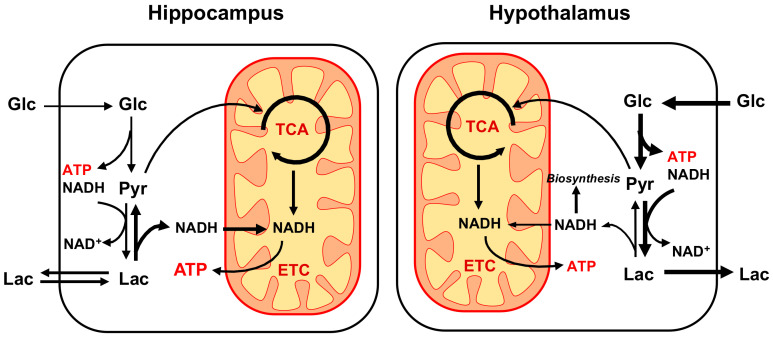
Schematic representation of main energy metabolic differences observed between hippocampus and hypothalamus using ^1^H-[^13^C]-MRS. Overall, energy production rate appears to be higher in hypothalamus (hippo: 20.5 vs. hypo: 23.7 μmol ATP/g/min) with a distinct metabolic organization. While tricarboxylic acid (TCA) cycle is comparable between structures, suggesting similar mitochondrial NADH production and oxidation, hippocampus relies significantly less on glycolysis to produce energy. The resulting lack of cytoplasmic ATP and NADH produced in the hippocampus are likely to be compensated by blood lactate influx that can provide NADH to be oxidized by the electron transport chain (ETC) in a process involving the malate-aspartate shuttle (MAS). As a result, the glycolysis-based metabolism of hypothalamus might produce more lactate and cytoplasmic NADH that is available for biosynthetic purpose.

**Table 1 metabolites-11-00050-t001:** Summary of hippocampal and hypothalamic metabolic fluxes assessed with a 1-compartment model of brain metabolism.

Flux	Hippocampus	Hypothalamus
V_TCA_	0.75 ± 0.05	0.87 ± 0.11
V_x_	0.50 ± 0.08	0.20 ± 0.06 ****
V_NT_	0.14 ± 0.01	0.14 ± 0.05
V_dil_^in^	0.23 ± 0.06	0.00 ± 0.06 ****
V_dil_^out^	0.21 ± 0.02	0.30 ± 0.02 ****
V_dil_^g^	0.06 ± 0.01	0.00 ± 0.00 ****
V_GABA_	0.06 ± 0.01	0.09 ± 0.02 ****
V_PC_	0.005 ± 0.005	0.010 ± 0.010
V_ex_^i^	0.01 ± 0.01	0.02 ± 0.02
CMR_Glc_	0.29 ± 0.02	0.47 ± 0.06 ****

Metabolic fluxes included the mitochondrial tricarboxylic acid (TCA) cycle (V_TCA_), the transmitochondrial flux (V_x_), a neurotransmission flux (V_NT_), an inward (V_dil_^in^) and outward flux (V_dil_^out^) from blood lactate, a glial dilution flux (V_dil_^g^), a glutamate-GABA cycle (V_GABA_), a pyruvate carboxylase flux (V_PC_), a GABA dilution flux (V_ex_^i^) and the cerebral metabolic rate of glucose (CMR_Glc_). Values are reported as mean ± SD, and statistics were done with unpaired Student’s *t*-test with **** *p* < 0.00001.

## Data Availability

Data is contained within the article or Appendix A.

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
