# Peer review of "In Vivo Metabolism of [1,6-^13^C_2_]Glucose Reveals Distinct Neuroenergetic Functionality between Mouse Hippocampus and Hypothalamus"

_metabolites, 2021, doi:10.3390/metabo11010050_

Round 1

Reviewer 1 Report

This is a well-written paper but there are several problems with data interpretation. First of all both brain structures the hypothalamus and hippocampus are involved in quite different tasks. Whereas the hippocampus is the element of episodical memory, most of the hypothalamus is involved in the control of organismal metabolism and energy metabolism in particular. 

The organismal energy metabolism is permanently and extensively controlled at the level of the hypothalamus whereas the hippocampal activity is generally related to learning and memory processes. That is why you see so significant difference in the activity of both structures. You should discuss these problems. Also in future research, you may include experimental manipulation with learning.

Author Response

Response: We would like to thank the reviewer for this comment and we agree with the point raised. In fact, this difference in the tasks, in which these two structures are involved, is precisely how we interpret our measured metabolic differences (see “3.3 Biological functionality underlying regional differences in energy metabolism”, p.9). In particular, we highlight several cytoarchitecture or functional characteristics of these two structures that might explain the observed differences, such as the metabolic regulation properties of hypothalamus and in particular its glucose sensing characteristics. Importantly, we found that both structures have differences in their energy production for a similar neurotransmission activity (VNT). This suggests that it is not the neuronal activity that is different, but the way energy is produced in these structures to give rise to this ‘activity’.

We have clarified the discussion in paragraph 3.3 (p.9 line 12), to emphasize that the observed difference is not related to neuronal activity, but metabolic activity. “In particular, our results suggest that hypothalamus undergoes significantly more aerobic glycolysis than hippocampus for a similar neurotransmission activity”. Furthermore, to clarify that we do not claim to have observed a difference in the neuroenergetic role (function) of both structures, which is already obvious, but a difference in how neuroenergetics are working (functionality), we have slightly changed the title of the article: we have replaced the work “function”, which can be misleading, for “functionality”. 

We appreciate the reviewer’s suggestion to include experimental manipulation with learning in future research.

Reviewer 2 Report

The authors aim to ascertain the metabolic differences between the hypothalamus and hippocampus using C13 labeled glucose in mice. The results are important and build upon previous reports examining metabolic differences in brain regions. Additional insight is provided about mitochondrial vs glycolytic ATP production.

General comments:

1) While the introduction provides a unfocused and brief rationale of why hippocampus and hypothalamus were selected, why were other regions not included - frontal cortex etc?

2) Despite no significance for GSH, it would be nice to see the concentration observed in the brain the supplement of these other metrics. GSH and other antioxidant cycles are heavily dependent on NADPH derived from glucose from the ppp. 

3) Why only male mice? Sex often alters metabolic rates. 

The authors clearly understand the limitations of their studies and provide possibilities for differences in studies including age and isoflurane use (which are likely contributors). 

Author Response

General comments:

1) While the introduction provides a unfocused and brief rationale of why hippocampus and hypothalamus were selected, why were other regions not included - frontal cortex etc?

Response: We agree with the reviewer that inclusion of other brain structures would allow a more systemic analysis of brain regional metabolic differences. Unfortunately, given the current technical challenges that are faced when using in vivo 13C-MRS, addition of more brain structures would have been beyond the scope of this study, which primarily aimed at using a similar acquisition and analysis protocol to compare two brain structures. As hypothalamus and hippocampus were studied previously with separate experimental protocols, we wanted to be able to compare our results with this previous work. Furthermore, since these two regions have high sensitivity to metabolic challenges, such as stress, we had particular interest in focusing on those ones as well. Nevertheless, we agree that including prefrontal cortex will certainly be important in future studies. To emphasize the rational for using only hippocampus and hypothalamus, we have now added the following sentence in the introduction (p.2, lines 16-18) “ Besides allowing a comparison with previous work that looked at both structures separately, hippocampus and hypothalamus were selected due to their distinct roles within the limbic system and their known sensitivity to metabolic challenges.”

2) Despite no significance for GSH, it would be nice to see the concentration observed in the brain the supplement of these other metrics. GSH and other antioxidant cycles are heavily dependent on NADPH derived from glucose from the ppp. 

Response: We agree with the reviewer that it is surprising not to observe differences in GSH between the two structures, given their implication in the pentose phosphate pathway (ppp). However, higher production rates of NADPH would not necessarily mean higher GSH levels, but most likely higher GSH-GSSG turnover. As our analysis cannot assess the ratio of GSH tot GSSG, nor the ppp in particular, the interpretation of the GSH levels remain limited within the scope of this study. Nevertheless, it is notable that higher NADPH production rates in hypothalamus might explain the ~2 fold drop in ascorbate concentration in that structure, suggesting high production of reducing equivalents diminishes the need for storing high level of antioxidants in the cells. This important point has been now added in the discussion: (p.10, lines 1-3) “Furthermore, higher reducing equivalent production rates in hypothalamus could potentially explain the relatively low levels of ascorbate antioxidant concentration in that structure (but similar GSH) compared to hippocampus.”

3) Why only male mice? Sex often alters metabolic rates. 

Response: We agree with the reviewer that inclusion of females adds important value to the findings. However, as this would have doubled the amount of data to collect and would be difficult to interpret with regard to previous literature focusing primarily on males, we have not oriented our study towards a sex-comparison of brain metabolism.

The authors clearly understand the limitations of their studies and provide possibilities for differences in studies including age and isoflurane use (which are likely contributors). 

Response: We would like to thank the reviewer for the positive and useful comments.